# NEURAL NETWORKS FOR IRREGULARLY OBSERVED CONTINUOUS-TIME STOCHASTIC PROCESSES

## ABSTRACT

Designing neural networks for continuous-time stochastic processes is challenging, especially when observations are made irregularly. In this article, we analyze neural networks from a frame theoretic perspective to identify the sufficient conditions that enable smoothly recoverable representations of signals in $L^2(\mathbb{R})$. Moreover, we show that, under certain assumptions, these properties hold even when signals are irregularly observed. As we obtain a family of (convolutional) neural networks that satisfy these conditions, we show that we can optimize our convolution filters while constraining them so that they effectively compute a Discrete Wavelet Transform. Such a neural network can efficiently divide the time-axis of a signal into orthogonal sub-spaces of different temporal scale and localization. We evaluate the resulting neural network on an assortment of synthetic and real-world tasks: parsimonious auto-encoding, video classification, and financial forecasting.

## INTRODUCTION

The predominant assumption made in deep learning for time series analysis is that observations are made regularly, with the same duration of time separating each successive timestamps (Cho et al., 2014; Graves et al., 2013; Sutskever et al., 2014; LeCun & Bengio, 1995; van den Oord et al., 2016; Bakshi & Stephanopoulos, 1993). However, this assumption is often inappropriate, as many real-world time series are observed irregularly and are, occasionally, event-driven (e.g., financial data, social networks, internet-of-things).

One common approach in working with irregularly observed time series is to interpolate the observations to realign them to a regular time-grid. However, interpolation schemes may result in spurious statistical artifacts, as shown in (Huth & Abergel, 2014; Belletti et al., 2017). Fortunately, procedures for working with irregularly observed time series in their unaltered form have been devised, notably in the field of Gaussian-processes and kernel-learning (Huth & Abergel, 2014; Belletti et al., 2017) and more recently in deep learning (Neil et al., 2016).

In this article, we investigate the underlying representation of time series data as it is processed by a neural network. Our objective is to identify a class of neural networks that provably guarantee information preservation for certain irregularly observed signals. In doing so, we must analyze neural networks from a frame theoretic perspective, which has enabled a clear understanding of the impact discrete sampling has on representations of continuous-time signals (Benedetto & Heller, 1990; Benedetto, 1992; Benedetto & Ferreira, 2003; Feichtinger & Gröchenig, 1994; Gröchenig, 1992; Mallat, 2008).

Although frame theory has historically been studied in the linear setting, recent work by Sun & Tang (2017) has related frames with non-linear operators in Banach space, to what can be interpreted as non-linear frames. Here, we extend this generalization of frames to characterize entire families of neural networks. In doing so, we can show that the composition of certain non-linear neural layers (i.e., convolutions and fully-connected layers) form non-linear frames in $L^2(\mathbb{R})$, while others do not (i.e., recurrent layers).

Moreover, frame theory can be used to analyze randomly-observed time series. In particular, when observations are made according to a family of self-exciting point processes known as Hawkes processes (Daley & Vere-Jones, 2007). We prove that such processes, under certain assumptions of stability, *almost surely* yield non-linear frames on a class of band-limited functions. That is to say,

that despite having discrete and irregular observations, the signal of interest can still be smoothly recovered.

As we obtain a family of convolutional neural networks that constitute non-linear frames, we show that under certain conditions, such networks can efficiently divide the time-axis of a time series into orthogonal sub-spaces of different temporal scale and localization. Namely, we optimize the weights of our convolution filters while constraining them so that they effectively compute a Discrete Wavelet Transform (Mallat, 1989). Our numerical experiments on synthetic data highlight this unique capacity that allows neural networks to learn sparse representations of signals in $L^2(\mathbb{R})$, and how such a property is particularly powerful when training parsimoniously parameterized auto-encoders. Such auto-encoders learn optimal ways of compressing certain classes of input signals.

Finally, we show that the ability of these networks to divide time series into a set sub-spaces, corresponding to different temporal scales and localization, can be composed with existing predictive frameworks to improve both accuracy and efficiency. This is demonstrated on real-world video classification and financial forecasting tasks.

CONTRIBUTIONS & ORGANIZATION

1. **Neural representations of $L^2(\mathbb{R})$:** We introduce the article with a theoretical analysis of the sufficient conditions on neural networks that enable smoothly recoverable representations of signals in $L^2(\mathbb{R})$ and prove that, under certain assumptions, this property holds true in the irregularly observed setting.

2. **Orthogonal representations in time:** We proceed to show that by enforcing certain constraints on convolutional filters, we can guarantee that the representation that the neural network produces only depends on the coordinates of the input signal in an learned orthonormal basis.

3. **Numerical experiments:** Finally, we evaluate the resulting constrained convolutional neural network on an assortment of synthetic and real-world tasks: parsimonious auto-encoding, video classification, and financial forecasting.

NOTATION

- $L^2(\mathbb{R})$ is the space of square-integrable real-valued functions defined on $\mathbb{R}$ and equipped with the norm induced by the inner product $f, g \in L^2(\mathbb{R}) \to \int_{t \in \mathbb{R}} f(t)g(t)dt$.

- $L^2_d(\mathbb{R})$ is the space of square-integrable d-dimensional vector-valued functions defined on $\mathbb{R}$ and equipped with the norm induced by the inner product $f, g \in L^2_d(\mathbb{R}) \to \int_{t \in \mathbb{R}} f(t)^T g(t)dt$.

- $l^2(\mathbb{Z})$ is the space of square-integrable real-valued sequences indexed by $\mathbb{Z}$ and equipped with the norm induced by the inner product $(x), (y) \in l^2(\mathbb{Z}) \to \sum_{n \in \mathbb{Z}} x_n y_n$.

- $l^2_d(\mathbb{Z})$ is the space of square-integrable d-dimensional vector-valued sequences indexed by $\mathbb{Z}$ and equipped with the norm induced by the inner product $(x), (y) \in l^2_d(\mathbb{Z}) \to \sum_{n \in \mathbb{Z}} x_n^T y_n$.

Note that the inner products of the spaces we consider are Hilbert spaces on the classes of equivalent functions for the Lebesgue measure.

- $FT[\cdot]$ denotes the Fourier transform and $\overline{z}$ denotes the complex conjugate of $z \in \mathbb{C}$. Recall the Fourier transform of a sequence $(x_n) \in \mathbb{Z}$ is given at any frequency $\omega$ by $FT[(x_n)](\omega) = \sum_{n \in \mathbb{Z}} e^{-2\pi i \omega n} x_n$.

- For a function $f : t \in \mathbb{R} \to f(t) \in \mathbb{R}^d$, $f(\cdot - h)$ is short-hand to denote $f : t \in \mathbb{R} \to f(t - h) \in \mathbb{R}^d$, likewise, $f(\cdot/\sigma)$ is used to denote $f : t \in \mathbb{R} \to f(t/\sigma) \in \mathbb{R}^d$.

- For a set $\mathcal{A}$, $(\xi)$ denotes the sequence $(\xi_n) \in \mathcal{A}^{\mathbb{Z}}$.

- $(\xi)[:: 2]$ denotes $(\xi_{2n})_{n \in \mathbb{Z}}$. That is, the dilation of a sequence by a factor of 2.

- For two vectors $(w_0, w_1) \in \mathbb{R}^{d_1} \times \mathbb{R}^{d_2}$, their concatenation is denoted by $[w_0, w_1] \in \mathbb{R}^{d_1 + d_2}$.

# 1 HOMEOMORPHIC NON-LINEAR ENCODINGS OF CONTINUOUS-TIME SIGNALS

We begin by investigating sufficient conditions on composite functions that guarantee such functions produce discretized representations of continuous-time signals that can be smoothly reconstructed.

## 1.1 FRAMES

To do so, we must leverage frame theory (Benedetto & Ferreira, 2003), a theory developed to precisely to characterize the suitable properties for linear representations of irregularly observed signals. Intuitively, a frame is a representation of a signal that enables signal recovery in a smooth manner (i.e., suitable for the representation to be homeomorphic).

Formally (Benedetto & Ferreira, 2003), we define a frame as an operator from $L^2(\mathbb{R})$ to $l^2(\mathbb{Z})$ that is characterized by a family of functions $(S_n)_{n \in \mathbb{Z}}$ in $L^2(\mathbb{R})$ (i.e., the *atoms* of the frame).

**Definition 1.1** *Linear frame of* $L^2(\mathbb{R})$*: A linear operator corresponding to the family* $(S) \in L^2(\mathbb{R})^{\mathbb{Z}}$,

$$\mathcal{F}_{(S)} : f \in L^2(\mathbb{R}) \to (<f, S_n>)_{n \in \mathbb{Z}} \in l^2(\mathbb{Z})$$

*is a frame of* $L^2(\mathbb{R})$ *if and only if there exist two real-valued constants* $0 < A \leq B$ *such that*

$$\forall f \in L^2(\mathbb{R}), \; A\|f\|_2^2 \leq \|\mathcal{F}_{(S)}(f)\|_2^2 \leq B\|f\|_2^2.$$

Representations provided by frames depend smoothly on their inputs. Moreover, a direct consequence of the definition above is that a frame is invertible in a smooth manner on its image.

There are many examples of frames. For now, we provide two concrete examples from (Benedetto & Ferreira, 2003; Mallat, 2008). Recall the definition of the Haar function as $t \in \mathbb{R} \to W_{\text{Haar}}(t) = 1$ if $t \in [0, 1/2)$, $-1$ if $t \in [1/2, 1)$, $0$ otherwise, the set of dilations and translations of $W_{\text{Haar}}$, namely $\left(W_{\text{Haar}}\left(\frac{\cdot - 2^l t}{2^l}\right)\right)_{t, l \in \mathbb{Z}}$ constitutes a frame of $L^2(\mathbb{R})$. Shannon's sampling theorem shows that, with sampling frequency $\frac{1}{\Delta t}$, the set of translated functions $(\sqrt{\Delta t} \frac{\sin(\pi(t-n)/\Delta t)}{\pi t})_{n \in \mathbb{Z}}$ is also a frame for the set of functions in $L^2(\mathbb{R})$ whose Fourier transforms are supported on the interval $\left[-\frac{1}{\Delta t}, \frac{1}{\Delta t}\right]$. In both cases, the atoms of the frame are orthonormal families of functions – it is trivial to prove that $A = B = 1$. While the first frame works for the entire space of square integrable functions, the second only applies to the sub-space of band-limited signals.

**Proposition 1.1** *Left inverses of Frames: If* $\mathcal{F}_{(S)}$ *is a frame of* $L^2(\mathbb{R})$ *then* $\mathcal{F}_{(S)}$ *has a left inverse* $\mathcal{F}^+ = (\mathcal{F}_{(S)}^* \mathcal{F}_{(S)})^{-1} \mathcal{F}_{(S)}^*$ *that is* $\frac{1}{\sqrt{A}}$ *Lipschitz, where* $\mathcal{F}_{(S)}^*$ *is the adjoint of* $\mathcal{F}_{(S)}$.

This fundamental proposition is proven in (Benedetto & Ferreira, 2003; Mallat, 2008). As our goal is to find the conditions for non-linear representations of $L^2(\mathbb{R})$ to be homeomorphic, we unfortunately can not leverage properties in the linear setting. Therefore, we must adopt an alternative definition. Let a non-linear frame be an operator from $L^2(\mathbb{R})$ to $l^2(\mathbb{Z})$ that is characterized by a family of functions $(S_n)_{n \in \mathbb{Z}}$ in $L^2(\mathbb{R})$ and a family of non-linear real valued functions $(\psi_n)_{n \in \mathbb{Z}}$ defined over $\mathbb{R}$ (or $\mathbb{R}^d$ as a trivial extension).

**Definition 1.2** *Non-linear frame of* $L^2(\mathbb{R})$*: A non-linear discrete representation scheme*

$$\mathcal{F}_{(S),(\psi)} : f \in L^2(\mathbb{R}) \to (\psi_n(<f, S_n>))_n \in l^2(\mathbb{Z})$$

*is a non-linear frame of* $L^2(\mathbb{R})$ *if there exist two real-valued constants* $0 < A \leq B$ *such that*

$$\forall f, g \in L^2(\mathbb{R}), \; A\|f - g\|_2^2 \leq \|\mathcal{F}_{(S),(\psi)}(f) - \mathcal{F}_{(S),(\psi)}(g)\|_2^2 \leq B\|f - g\|_2^2.$$

It is worth noting that a linear frame (in the standard definition of the term) is still a frame in this non-linear setting.

**Proposition 1.2** *Smoothness of signal recovery: A non-linear frame is invertible on its image of* $L^2(\mathbb{R})$ *and the inverse is* $\frac{1}{\sqrt{A}}$ *Lipschitz.*

**Proof 1.1** *Consider $f, g \in L^2(\mathbb{R})$ such that $\mathcal{F}_{(S),(\psi)}(f) = \mathcal{F}_{(S),(\psi)}(g)$, then, as $A > 0$, $\|f - g\|_2 = 0$ and therefore $f$ and $g$ are in same equivalence class in $L^2(\mathbb{R})$. Therefore, $\mathcal{F}_{(S),(\psi)}$ is injective and if we consider $(x), (y) \in \mathcal{F}_{(S),(\psi)}(L^2(\mathbb{R}))$ and if we denote by $f_x$ the only element in $L^2(\mathbb{R})$ such that $\mathcal{F}_{(S),(\psi)}(f_x) = (x)$, and $f_y$ the only element in $L^2(\mathbb{R})$ such that $\mathcal{F}_{(S),(\psi)}(f_y) = (y)$ then, by definition of a non-linear frame, $\|f_x - f_y\|_2^2 \leq \frac{1}{A}\|x - y\|_2^2$.* ∎

In a later section, we will show that smooth signal recovery is crucial for non-linear signal approximations (consisting of a finite number of coefficients) to remain stable during reconstruction. However in order to show this, we must first explore the sufficient conditions on non-linear operators to produce non-linear frames. We start by introducing several definitions on multivariate real-valued functions.

**Definition 1.3** *Bi-Lipschitz-Invertible (BLI) operators: An operator $\Phi : l^2(\mathbb{Z}) \rightarrow l^2(\mathbb{Z})$ such that*

$$\exists \, 0 < A < B \; s.t. \; \forall (x_n), (y_n) \in L^2(\mathbb{Z}), \; A\|x - y\|_2^2 \leq \|\Phi(x) - \Phi(y)\|_2^2 \leq B\|x - y\|_2^2,$$

*is a BLI operator and we refer to $(A, B)$ as some **framing constants** of $\Phi$.*

**Theorem 1.1** *BLI operators and linear frames: Let $(\Phi_l)_{l=1...L}$ be a collection of BLI operators with framing constants $((A_l, B_l))_{l=1...L}$ and $\mathcal{F}$ a frame on $L^2(\mathbb{R})$ with framing constants $A_0$ and $B_0$. The composite operator $\phi_L \circ \cdots \circ \phi_1 \circ \mathcal{F}$ is a non-linear frame of $L_2(\mathbb{R})$ with framing constants $\prod_{l=0}^{L} A_0$ and $\prod_{l=0}^{L} B_0$.*

**Proof 1.2** *The proof of the theorem is immediate but we use it to expose how our careful choice of the definition of non-linear frames is leveraged. First let us recall that injectivity is preserved by composition. Then, we initiate an immediate proof by induction with a simple remark: consider two functions $f, g \in L^2(\mathbb{R})$*

$$A_1 A_0 \|f - g\|_2^2 \leq A_1 \|\mathcal{F}(f) - \mathcal{F}(g)\|^2$$
$$\leq B_1 \|\mathcal{F}(f) - \mathcal{F}(g)\|_2^2$$
$$\leq B_1 B_0 \|f - g\|_2^2$$

*To conclude the proof, a similar statement can then be made if we compose $\Phi_1 \circ \mathcal{F}$ by $\Phi_2, \ldots \Phi_L$.* ∎

This proof allows us to make guarantees about operator pipelines while relying on conditions that are simple to verify. We can now use the theory we have established to analyze the representational properties of neural networks; in particular, convolutional neural networks (CNN) and recurrent neural networks (RNN).

## 1.2 Sufficient conditions on CNNs to produce non-linear frames

Here, we study representational properties of recent CNN architectures (Chollet, 2016; Kaiser et al., 2017; Lebedev et al., 2014) that rely on depth-wise separable convolutions. We show that by enforcing certain constraints on the structure of temporal filters, we obtain a network that is, provably, a non-linear frame. Here we trade off expressiveness for representational guarantees as we impose constraints on network parameters.

In depth-wise separated convolution stacks (Chollet, 2016; Kaiser et al., 2017) a temporal convolution is applied before a depth-wise linear transform and finally a leaky ReLU layer. We assume that the depth-wise linear operators being learned are all full rank (or full column rank if they increase the number of dimensions of the representation). Such an assumption makes sense for CNNs being trained by a stochastic optimization method with non-pathological data-sets.

Inspired by the multi-scale parsing enabled by the discrete wavelet transform or dyadic wavelet transform we employ time domain convolutions that are conjugate mirror filters (Mallat, 2008). Such time domain filters constitute a decomposition filter bank consisting of cascading convolutions. The decomposition filter bank admits a dual reconstruction filter bank thereby guaranteeing injectivity.

**Definition 1.4** *Element-wise Leaky ReLU (LReLU): Consider $0 < \alpha << 1$, LReLU applies a piece-wise linear function element-wise as*

$$LReLU_\alpha : (x_n) \in l_d^2(\mathbb{Z}) \rightarrow (\max(\alpha x_n, x_n))_{n \in \mathbb{Z}} \in l_d^2(\mathbb{Z}).$$

**Definition 1.5** *Depth-wise fully connected layer (DFC): Consider two integers $d_i$ and $d_o$, $A \in \mathbb{R}^{d_o,d_i}$ and $b \in \mathbb{R}^{d_o}$*

$$DFC_{A,b}: (x_n) \in l^2_{d_i}(\mathbb{Z}) \to (LReLU(A \times x_n + b))_{n \in \mathbb{Z}} \in l^2_{d_o}(\mathbb{Z}).$$

**Lemma 1.1** *Full column rank DFC (FDFC) layers are BLI: The function*

$$w \in \mathbb{R}^{d_i} \to LReLU(Aw + b) \in \mathbb{R}^{d_o}$$

*with $d_i \leq d_o$ and $A$ is full column rank is left invertible. Also, the left inverse is Lipschitz as $\exists 0 < m \leq M \in \mathbb{R}$ such that*

$$\forall w \in \mathbb{R}^d_i, \ m\|w - w'\|^2 \leq \|LReLU(Aw + b) - LReLU(Aw' + b)\|^2 \leq M\|w - w'\|^2.$$

**Proof 1.3** *As LReLU is strictly increasing and continuous therefore it is invertible and as $A$ is full column rank it admits a left inverse which proves the first part of the lemma. We finish the proof by using the fact that linear functions in vector spaces of finite dimensions are Lipschitz, the fact that LReLU and its inverse are Lipschitz, and the fact that Lipschitz-ness is preserved by composition.* ∎

Let us now study the representational properties of time domain convolution layers whose filters are constrained in the Frequency domain.

**Definition 1.6** *Reconstructible convolution layer (RConv): Consider two convolution filters $h, g \in l^2(\mathbb{Z})$ such that there exist $\widetilde{h}, \widetilde{g} \in l^2(\mathbb{Z})$ and*

$$\forall \omega \in \mathbb{R}, FT[\widetilde{h}] \times \overline{FT[h]}(\omega) + FT[\widetilde{g}] \times \overline{FT[g]}](\omega) = 2. \tag{1}$$

*The following convolution is a Reconstructible convolution layer:*

$$RConv_{l,h,g}: (x_n) \in l^2_d(\mathbb{Z}) \to ([h * x_n, g * x_n])_{n \in \mathbb{Z}} \in l^2_{2 \times d}(\mathbb{Z}).$$

Later on, we show that entire families of such $\widetilde{h}, \widetilde{g} \in l^2(\mathbb{Z})$ exist under some conditions on $h, g$. In particular Eq. (4) will provide simple sufficient conditions on $h$ and $g$ for Eq. (1) to hold.

**Lemma 1.2** *Temporal convolutions allowing reconstruction: Consider four temporal convolution filters $h, \tilde{h}, g, \tilde{g}$ such that their Fourier transforms satisfy* (1). *If $x^0_{l+1} = h * x_l$ and $x^1_{l+1} = g * x_l$ then $x_l = \frac{1}{2}\left(\widetilde{h}[:: -1] * x^0_{l+1} + \widetilde{g}[:: -1] * x^1_{l+1}\right)$ (where $[:: -1]$ means that we iterate in reverse order on the filter weights) which proves the pair of convolution filters constitutes an invertible operator. Also, $\exists \, 0 < m \leq M$ such that*

$$m\|x_l\|^2_2 \leq \|x^0_{l+1}\|^2_2 + \|x^1_{l+1}\|^2_2 \leq M\|x_l\|^2_2.$$

**Proof 1.4** *By definition $\widetilde{h}[:: -1] * x^0_{l+1} + \widetilde{g}[:: -1] * x^1_{l+1} = \widetilde{h}[:: -1] * h * x^l + \tilde{g}[:: -1] * g * x^l$, therefore, if we recall that the Fourier Transform diagonalizes convolutions and turns time reversal into complex conjugacy, we have*

$$FT\left[\widetilde{h}[:: -1] * x^0_{l+1} + \widetilde{g}[:: -1] * x^1_{l+1}\right] = \left(\overline{FT\left[\widetilde{h}\right]} \times FT[h] + \overline{FT[\widetilde{g}]} \times FT[g]\right) \times FT\left[x^l\right] = 2FT\left[x^l\right]$$

*with condition* (1). *Also with the Plancherel formula $\|h * x^l\|^2_2 = \|FT[h * x^l]\|^2_2 \leq \|FT[h]\|^2_2 \times \|FT[x^l]\|^2_2 = \|h\|^2_2\|x^l\|^2_2$. And finally, $\|FT[x^l]\|^2_2 = \frac{1}{4}\|FT[\widetilde{h}[:: -1] * x^0_{l+1} + \widetilde{g}[:: -1] * x^1_{l+1}]\|^2_2 \leq \frac{1}{4}\left(\|FT[\widetilde{h}[:: -1] * x^0_{l+1}]\|^2_2 + \|FT[\widetilde{g}[:: -1] * x^1_{l+1}]\|^2_2\right)$ which concludes the proof with*

$$m = \frac{\|\widetilde{h}\|^2_2 + \|\widetilde{g}\|^2_2}{4} \text{ and } M = \|h\|^2_2 + \|g\|^2_2. \blacksquare$$

We represent the recomposition operation introduced in Lemma 1.2 in . The next proposition shows how the convolutions RConv and non-linearity LReLU can be interleaved to produce non-linear frames.

**Proposition 1.3** *Compositions of FDFC and RConv layers are BLI: The composite function from $l^2_{d_i}(\mathbb{Z})$ onto $l^2_{d_o}(\mathbb{Z})$*

$$FDFC_{A_L,b_L} \circ RConv_{L,h_l,g_l} \circ \cdots \circ FDFC_{A_1,b_1} \circ RConv_{1,h_1,g_1}$$

*is BLI.*

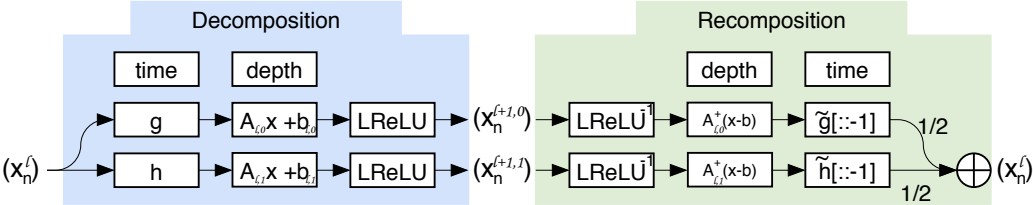

Figure 1: Decomposition and recomposition principle for depth-wise separable convolutions. We denote the Moore-Penrose inverse of the full column rank matrix $A$ by $A^+$.

**Proof 1.5** *Let us recall again that injectivity is stable by composition of operators. It is also clear that non-linear framing conditions remain true as composite bi-Lipschitz functions are also bi-Lipschitz.* ∎

With the proposition above it is now trivial to prove the theorem below.

**Theorem 1.2** *Compositions of FDFC and dilated RConv layers are non-linear frame:* *If $\mathcal{F}$ is a frame of $L^2(\mathbb{R})$ then the composite function from $L^2_{d_i}(\mathbb{R})$ onto $l^2_{d_o}(\mathbb{Z})$*

$$FDFC_{A_L, b_L} \circ RConv_{L, h_L, g_L} \circ \cdots \circ FDFC_{A_1, b_1} \circ RConv_{1, h_1, g_1} \circ \mathcal{F}$$

*is a non-linear frame.*

Now, we expose the framing properties of RNNs (for an introduction on RNNs we refer the reader to (Bengio et al., 1994)). For the vast majority of popular recurrent architectures (for instance, LSTMs, GRUs (Hochreiter & Schmidhuber, 1997; Cho et al., 2014)) the use of bounded output layers leads to saturation and vanishing gradients. With such vanishing gradients Bengio et al. (1993), it is possible to find series of input sequences that diverge in $l^2(\mathbb{Z})$ while their outputs through the RNN are a Cauchy sequence.

**Proposition 1.4** *Saturating RNNs do not provide non-linear frames:* *Let us consider a RNN*

$$\mathcal{F}_{(S),(\psi)} : f \in L^2(\mathbb{R}) \to (\psi_n((< f, S_{n'} >)_{n' \in \mathbb{Z}}))_n \in l^2(\mathbb{Z})$$

*where $\mathcal{F}_{(S)}$ is a given linear frame of $L^2(\mathbb{R})$. If there exists a sequence $(v^k)_{k \in \mathbb{Z}} \in \mathcal{F}_{(S)}(L^2(\mathbb{R}))^{\mathbb{Z}}$ such that $((\psi_n(v^k))_{n \in \mathbb{Z}})_{k \in \mathbb{N}}$ is a Cauchy sequence in $l^2(\mathbb{Z})$ while $\|v^k\|_2 \to +\infty$, then $\mathcal{F}_{(S),(\psi)}$ is not a linear frame.*

**Proof 1.6** *With the assumption on $(v^k)_{k \in \mathbb{Z}} \in \mathcal{F}_{(S)}(L^2(\mathbb{R}))^{\mathbb{Z}}$, there cannot exist $A > 0$ such that*

$$\forall k_0, k_1 \in \mathbb{Z}, \ A\|v^{k_0} - v^{k_1}\|_2^2 \le \|(\psi_n((< v^{k_0}, S_{n'} >)_{n' \in \mathbb{Z}}))_{n \in \mathbb{Z}} - (\psi_n((< v^{k_1}, S_{n'} >)_{n' \in \mathbb{Z}}))_{n \in \mathbb{Z}}\|_2^2$$

*as the sequence $(v^k)$ would then be Cauchy and therefore converge as $l^2(\mathbb{Z})$ is complete for the $l^2$ norm.* ∎

Such a proposition highlights a key difference between the representational ability of RNNs and CNNs. We explore representations of irregularly sampled data through the lens of non-linear frames.

## 1.3    IRREGULAR SAMPLING

We now show that even when signals are irregularly observed by a random sampling process, that particular neural networks can still, *almost surely* produce a homeomorphic representation.

Sampling by Hawkes processes is a common assumption in finance, seismology, and social media analytics (Ogata, 1988; Bacry et al., 2015; Belletti et al., 2017; Yang & Zha, 2013; Li & Zha, 2013; Zhao et al., 2015). We use $(\mathcal{I}_t^N)$ to denote the canonical filtration associated with the stochastic process $(N_t)_{t \in \mathbb{R}}$. We recommend (Daley & Vere-Jones, 2007) for a more thorough introduction to this concept. As a simplification, we denote $\mathcal{I}_t^N$ to be the information generated by $(N_s)_{s < t}$.

**Definition 1.7** *Random sampling Hawkes process: A stochastic point process $(N_t)$ (Daley & Vere-Jones, 2007) defines a random measure over the axis of time and is defined by stochastic local Poisson intensities for $(N_t)$*

$$\lambda_t = lim_{dt \to 0+} \frac{E\left[N(t + dt) - N(t)|\mathcal{I}_t^N\right]}{dt}.$$

*For a Hawkes process characterized by $\phi : t \in \mathbb{R} \to \phi(t) \geq 0, \ \forall t < 0, \phi(t) = 0, \mu \geq 0$, we assume*

$$\lambda_t = \mu + \phi * dN_t. \tag{2}$$

In other words, $\lambda_t$ is the number of observations per unit of time expected given the events that occurred until time $t$. Intuitively, if $\lambda_t$ is higher, then it is more likely for observations to be available shortly after the time $t$.

As in (Belletti et al., 2017; Bacry et al., 2015), Hawkes processes can be used to model the random observation time of a continuous-time series in a setting where information is observed asynchronously in an event-driven manner across multiple channels (the extension to multi-variate point processes is immediate).

**Proposition 1.5** *Sampling density of stable Hawkes processes: If the Hawkes process* (2) *is stable (i.e. $\int_{t \in \mathbb{R}} \phi(t)dt < 1$), then almost surely*

$$\lim_{t \to +\infty} \frac{N_t}{t} = \frac{\mu}{1 - \int_{t \in \mathbb{R}} \phi(t)dt}.$$

A complete proof of the ergodic behavior of stable Hawkes processes is provided in (Daley & Vere-Jones, 2007). Now, given an asymptotic Nyquist sampling rate for a random sampling scheme, the following lemma delineates which frames can still be used for signal recovery. In particular, we can no longer recover all signals in an unambiguous manner. Hence, exact recovery is only possible for band-limited functions (i.e. functions whose Fourier transform has bounded support).

**Theorem 1.3** *Irregular sampling frames: If a sequence of time-stamps have the property that*

$$\lim_{n \to \pm\infty} t_n = \pm\infty \ and \ \lim_{n \to +\infty} \frac{n}{t_n} > 2R_1,$$

*then $(e^{-2it_n \cdot})$ is a frame $\mathcal{F}$ of $L^2[-R, R]$ (where the real axis represents sampling frequencies) with left inverse $\mathcal{F}^+$. Considering $R < R_1$ and $S \in L^2(\mathbb{R})$ such that*

$$\|FT[S]\|_\infty < +\infty, \ support(FT[S]) \subset [-R_1, R_1] \ and \ \forall \omega \in [-R, R], \ FT[S](\omega) = 1, \tag{3}$$

*then*

$$\forall f \in L^2(\mathbb{R}) \ s.t. \ support(FT[f]) \subset [-R, R], f = \sum x_n(f)S(\cdot - t_n)$$

*where $x_n(f) = \int_{\omega = -R_1}^{R_1} \mathcal{F}^+(FT[f]1_{[-R_1, R_1]})(\omega)e^{2it_n\omega}d\omega$.*

Complete proof is given in (Benedetto & Heller, 1990; Benedetto, 1992); the theorem is regarded as the fundamental theorem of frame analysis for irregularly observed signals. We now leverage the fundamental properties that were obtained in the deterministic setting and extend them to provide guarantees under random sampling schemes.

**Proposition 1.6** *Under Hawkes process random sampling, framing is preserved almost surely: Let $(t_n)_{n \in \mathbb{Z}}$ be a family of sampling time-stamps generated by a stable Hawkes process whose intensity follows the dynamics described in* (2), *denote*

$$R = \frac{1}{2} \frac{\mu}{1 - \int_{t \in \mathbb{R}} \phi(t)dt},$$

*and let $S$ be a frame operator abiding by conditions* (3), *then almost surely the frame is injective on $\{f \in L^2(\mathbb{R}) \ s.t. \ support(FT[f]) \subset [-R, R]\}$ when translated by the irregular time-stamps $(t_n)_{n \in \mathbb{Z}}$.*

**Proof 1.7** *The proposition is a direct consequence of Prop. 1.5 and Theorem 1.3.* ∎

**Theorem 1.4** *Recovery of randomly observed band-limited signals:* *Let* $(t_n)_{n \in \mathbb{Z}}$ *be a family of sampling time-stamps generated by a stable Hawkes process whose intensity follows the dynamics described in (2) Consider* $R = \frac{1}{2} \frac{\mu}{1 - \int_{t \in \mathbb{R}} \phi(t) dt}$*, let* $\mathcal{F}_{(S)}$ *be a frame operator with atoms* $(S(\cdot - t_n))_{n \in \mathbb{Z}}$ *given by (3). A composite function from* $L^2_{d_i}(\mathbb{R})$ *onto* $l^2_{d_o}(\mathbb{Z})$

$$FDFC_{A_L, b_L} \circ RConv_{L, h_L, g_L} \circ \cdots \circ FDFC_{A_1, b_1} \circ RConv_{1, h_1, g_1} \circ \mathcal{F}_{(S(\cdot - t_n))_{n \in \mathbb{Z}}}$$

*is almost surely a non-linear frame over the set of functions in* $L^2(\mathbb{R})$ *whose Fourier Transform has its support included in* $[-R, R]$*. In particular such a representation is invertible on its image by a Lipschitz inverse.*

**Proof 1.8** *Previously, we proved Theorem 1.2 on the preservation of framing properties by composition with FDFC and RConv layers. In Prop. 1.6 we proved that* $\mathcal{F}_{(S(\cdot - t_n))_{n \in \mathbb{Z}}}$ *is almost surely a frame of the subset of* $L^2(\mathbb{R})$ *of functions with band-limit* $[-R, R]$*.* ∎

One concern, however, is that the theorems we developed assumed observations on the entire real axis are available as well as infinite representations indexed by $\mathbb{Z}$. In particular, a theory of framing for band-limited functions is useful but only applies to periodic functions. Bounded support function of $L^2(\mathbb{R})$ are part of the many examples that are not band-limited Mallat (2008); Benedetto & Ferreira (2003).

## 1.4 FINITE APPROXIMATIONS OF NON BAND-LIMITED SIGNALS

In our objective to develop theoretical statements that can be leveraged in practice (i.e., when computing with finite time and memory), we must now extend our analysis to (1) functions observed on compact intervals and (2) finite approximations of signals. The following statements show how the requirements of Lipschitz-ness in non-linear frames provide guarantees on the impact of approximation errors associated with finite representation of continuous-time signals.

The theorems above can be employed for irregularly observed functions that are periodic and band-limited (Benedetto & Ferreira, 2003; Benedetto, 1992; Mallat, 2008). However, since we hope to develop a representational theory that is applicable to non-stationary signals, we must also consider non-periodic functions. In the appendix, we show how wavelet decomposition can efficiently approximate certain classes of functions that are smooth and not band-limited.

With $\lfloor \log_2(N) \rfloor$ scales of decomposition and $O(N)$ scalars representing the approximation of $f$ as $P_W^{\lfloor \log_2(N) \rfloor}(f)$ the Wavelet approximation error is $O(2^{-N\alpha})$ in $L^2$ norm for the space of $\alpha$ Lipschitz functions (see Definition 4.1 in appendix). As we employ functions that are BLI the impact of the approximation error remains controlled.

**Proposition 1.7** *Approximate representation:* *Let* $\Phi_L \circ \cdots \circ \Phi_1$ *be a BLI function from* $l^2(\mathbb{Z})$ *onto* $l^2(\mathbb{Z})$*. Let* $P_W^l$ *be the projection operator from* $L^2([0, 1])$ *onto a Wavelet basis of* $l$ *scales,*

$$\|\Phi_L \circ \cdots \circ \Phi_1 \circ P_W^{\lfloor \log_2(N) \rfloor}(f) - \Phi_L \circ \cdots \circ \Phi_1 P_W^{+\infty}(f)\|_2^2 = O(2^{-N2\alpha}).$$

In other words, the numerical representation can be arbitrarily close to the true representation of smooth, continuous-time functions with compact support. Indeed, if $W$ is a wavelet basis, then $\text{Proj}_W^{+\infty}(f) = f$. The argument stresses the critical role of our assumption of the Lipschitz-ness of frames and the BLI functions which guarantees that representations based on approximations can be arbitrarily accurate.

## 2 ORTHOGONAL MULTI-RESOLUTION CONVOLUTIONS IN TIME

So far, we have focused on sufficient conditions to make accurate representation of continuous-time signals possible as they are observed randomly and as the corresponding observation are processed non-linearly. We now show that additional conditions on time-domain convolutional filter banks further guarantee that the representation is minimal (i.e., produces orthogonal outputs).

## 2.1 Multi-resolution representation

As our goal is to obtain different representations of a time series while avoiding redundancy, let us introduce multi-resolution approximations (Mallat, 2008).

**Definition 2.1** *Multi-resolution approximation: A family $(H_l)_{l \in \mathbb{Z}}$ of closed sub-spaces of $L^2(\mathbb{R})$ is a multi-resolution approximation if the family is **nested** ($\forall l \in \mathbb{Z}$, $H_{l+1} \subset H_l$); **dense** ($\overline{\cup_{l \in \mathbb{Z}} H_l} = L^2(\mathbb{R})$); **separated** ($\cap_{l \in \mathbb{Z}} H_l = \{0\}$); **causal** ($\forall l \in \mathbb{Z}, g(\cdot) \in H_l \Rightarrow g(\cdot/2) \in H_{l+1}$); **stable by trans-lation** ($\forall (t, l) \in \mathbb{Z}, g(\cdot) \in H_l \Rightarrow g(\cdot - 2^l t) \in H_l$). In addition we require that $(H_0)$ there exists an orthonormal family $(S(\cdot - n))_{n \in \mathbb{Z}}$ such that $span((S(\cdot)_{n \in \mathbb{Z}})) = H_l$, i.e. $S(\cdot - n)$ is a **Riesz basis** of $H_0$ with **scaling function** $S$.*

Such Riesz basis is proven to exist in (Mallat, 2008); the family of Haar wavelets is merely an example. General conditions for a function $S \in L^2(\mathbb{R})$ to be a scaling function are given by the following theorem.

**Theorem 2.1** *Conjugate mirror temporal convolution layer (CMConv): (Mallat, 1989) Let $\kappa_S$ and $\kappa_W$ in $l^2(\mathbb{Z})$ be two convolution filters such that*

$$FT[\kappa_S](0) = \sqrt{2}, \tag{4}$$

$$\forall \omega \in [-1/2, 1/2], \ |FT[\kappa_S](\omega)|^2 + |FT[\kappa_S](\omega + 1/2)|^2 = 2, \tag{5}$$

$$\kappa_{Wn} = (-1)^{1-n} \kappa_{S-n}. \tag{6}$$

We further assume that $\inf_{f \in [-\frac{1}{4}, \frac{1}{4}]} FT[\kappa_S](f) > 0$. The inverse Fourier Transform of $\omega \to \prod_{p=1}^{+\infty} \frac{FT[\kappa_S](2^{-p}\omega)}{\sqrt{2}}$ is a scaling function $S$ of $L^2(\mathbb{R})$ for a multi-resolution approximation. Moreover, the Wavelet function $W$ defined as the inverse Fourier transform of $\frac{1}{\sqrt{2}} FT[\kappa_W] FT[S]$ is such that for any scale $l \left\{ W_{l,n} = \frac{1}{\sqrt{2^l}} W\left(\frac{\cdot - 2^l n}{2^l}\right) | n \in \mathbb{Z} \right\}$ is an orthonormal basis $W_l$ defined as the orthogonal complement of $H_l$ in $H_{l+1}$. In particular, $(W_{l,n})_{l \in \mathbb{Z}, n \in \mathbb{Z}}$ is an orthonormal basis of $L^2(\mathbb{R})$.

## 2.2 Depth-wise separable convolutions and multi-resolution approximations

We now show how depth-wise separable convolutions with scaling and wavelet filters quickly come with guarantees of orthogonality.

In the following we consider an input space with $d$ input channels and a series of affine operators with increasing output dimensions $(d_l)_{l=1}^L$. We denote $\mathcal{F}_{S(\cdot - t_n)_{n \in \mathbb{Z}}}$ by $\mathcal{F}$ to simplify notations.

**Theorem 2.2** *Conjugate mirror convolutions and FDFC: Consider two convolution filters $\kappa_S$ and $\kappa_W$, if $\kappa_S, \kappa_W$ respect the conditions Eq. (4) and $\inf_{\omega \in [-\frac{1}{4}, \frac{1}{4}]} FT[\kappa_S](\omega) > 0$, then $\kappa_S, \kappa_W$ constitute a pair of RConv filters. Consider the function which to $f \in L_d^2(\mathbb{R})$ associates $(\theta_l(f))_{l=1...L}$:*

$$f \to (< f, S(\cdot - n) >)_{n \in \mathbb{Z}} \quad \overset{(\kappa_W * \cdot)[::2]}{\to} \quad \overset{BLI}{\to} \quad \cdots \quad \cdots \quad \overset{BLI}{\to} (\theta_1(f)_n)_{n \in \mathbb{Z}}$$
$$\searrow \quad \overset{(\kappa_W * \cdot)[::2]}{\to} \quad \overset{BLI}{\to} \quad \cdots \quad \overset{BLI}{\to} (\theta_2(f)_n)_{n \in \mathbb{Z}}$$
$$\overset{(\kappa_S * \cdot)[::2]}{\searrow} \quad \overset{BLI}{\to} \quad \cdots \quad \vdots$$
$$\overset{(\kappa_S * \cdot)[::2]}{\searrow} \quad \cdots \quad \ddots$$
$$\overset{BLI}{\to} (\theta_L(f)_n)_{n \in \mathbb{Z}}$$

The representation $(\theta_l(f))_{l=1...L}$ is a non-linear frame that only depends on the coordinates of $f$ in an orthonormal basis of $L^2(\mathbb{R})$.

**Proof 2.1** *We start the proof by showing that $(x_n) \in l^2(\mathbb{Z}) \to ([(\kappa_W * x)_{2n}, (\kappa_S * x)_{2n}]) \in l^2(\mathbb{Z})$ is a RConv reconstructible convolution layer. Based on Eq. (4), as $\kappa_{Wn} = (-1)^{1-n} \kappa_{-n}^S$ we have $FT[\kappa_W](\omega) = e^{i\pi\omega} \overline{FT[\kappa_S](\omega + \frac{1}{2})}$ and then as $\forall \omega \in [-1/2, 1/2], \ |FT[\kappa_S](\omega)|^2 + |FT[\kappa_S](\omega + $*

$1/2)|^2 = 2$ *the first part of the proof is concluded. The second part of the proof utilizes the fact that the cascading convolutions above compute a Discrete Wavelet Transform Mallat (2008). Therefore,* $(\theta_1(f)_n)_{n \in \mathbb{Z}} = (\theta_1(<f, W_{1,n}>)_n)_{n \in \mathbb{Z}}, (\theta_2(f)_n)_{n \in \mathbb{Z}} = (\theta_2(<f, W_{2,n}>)_n)_{n \in \mathbb{Z}}, \ldots, (\theta_{L-1}(f)_n)_{n \in \mathbb{Z}} = (\theta_{L-1}(<f, W_{L-1,n}>)_n)_{n \in \mathbb{Z}}$ *and* $(\theta_1(f)_n)_{n \in \mathbb{Z}} = (\theta_L(<f, S_{L-1,n}>)_n)_{n \in \mathbb{Z}}$ *where* $\left\{ S_{l,n} = \frac{1}{\sqrt{2^l}} S\left(\frac{-2^l n}{2^l}\right) | n \in \mathbb{Z} \right\}$. ∎

The cascaded time domain convolutions being computed yield the coordinates of $f$ in an orthonormal basis. Therefore, as the orthogonal CNN grows deeper it can only yield novel orthogonal information about the input signal that is informative of its properties on a particular temporal scale. Such is the nature of our efficiency claim for the neural networks we consider. A key point here is that the 1x1 convolutions operate in depth and not along the axis of time which preserves the temporal scaling properties of the Discrete Wavelet Transform.

## 2.3 TRAINING OF RCONV LAYERS BY ALTERNATING SGD STEPS AND DC PROGRAMMING:

As noted in Mallat (2008) the constraint $\inf_{\omega \in [-\frac{1}{4}, \frac{1}{4}]}, |FT[\kappa_S](\omega)| > 0$ is always met in practice, which our numerical experiments confirm. The two critical constraints on the design of the temporal convolution filters is (1) that $\kappa_W = (-1)^{1-n} \kappa_S[:: -1]$ and $FT[\kappa_S](0) = \sqrt{2}$, which is trivial to enforce, and (2) that $\forall \omega \in \mathbb{R}, |FT[\kappa_S](\omega)|^2 + |FT[\kappa_S](\omega + \frac{1}{2})|^2 = 2$.

In our implementation we approximate the constraint by computing the Fast Fourier Transform of the filter, since it is defined discretely in time by a finite set of weights. Therefore, we interleave the normal training step of $\kappa_S$ with solving the following following minimization problem

$$\min_{\kappa_S \in \mathbb{R}^H} \sum_{\omega = 0, \frac{1}{N-1}}^{1} \left| |FT[\kappa_S](\omega)|^2 + |FT[\kappa_S](\omega + \frac{1}{2})|^2 - 2 \right|$$

where $H$ is the number of free parameters we allow in our temporal convolution filter. Such an optimization problem can be rewritten as a difference of convex (DC) functions (as $|x| = \max(0, x) - \min(0, x)$, $\kappa_s \rightarrow |FT[\kappa_S](\omega)|^2 + |FT[\kappa_S](\omega + \frac{1}{2})|^2 - 2$ is clearly convex and convexity is stable by composition by a non-decreasing function) and an adapted solver (Tao & An, 1998) can then take advantage of the particular structure of the problem to find an optimal solution rapidly.

## 3 NUMERICAL EXPERIMENTS

Here, we show that the sufficient conditions for neural networks to yield non-linear frames are computationally tractable. The following experiments explore the empirical properties of such neural networks compared to various baselines.

### 3.1 SYNTHETIC COMPRESSION EXPERIMENT

In our first numerical experiment, we generate regularly sampled non-stationary stochastic processes, characterized by a random mixture of Gabor functions (Mallat, 2008) and step functions. As shown in Figure 2, the resulting signals are highly irregular, lack permanent seasonality, and have compact support. The objective here is to devise a procedure to train conjugate mirror (convolutional) filters with stochastic optimization methods to progressively improve representational power.

We train a 16 parameter filter $\kappa_S$ to optimally conduct the following compression (i.e., auto-encoding) task. The pair of filters specified in Eq. (4) are employed as in Theorem 2.2 to produce the coordinates of the input signal in the wavelet basis corresponding to the (learned) filters $\kappa_S$ and $\kappa_W$. The input signals are uni-variate with 128 observations each. The encoding, therefore, initially consists of 128 scalar values, of which, only the 64 with higher magnitude are selected – all other values are set to 0.

An inverse Discrete Wavelet Transform is then employed to reconstruct the input signal. The quality of this reconstruction is measured by the squared $L_2$ loss, which penalizes discrepancies between the input signal and its reconstruction. To train this model, we use a stochastic optimization algorithm, RMSProp, to minimize the aforementioned loss. We train for 2,500 iterations with a learning rate of $10^{-3}$. This optimization is interleaved with a constraint enforcing program that enforces Eq. (4) every

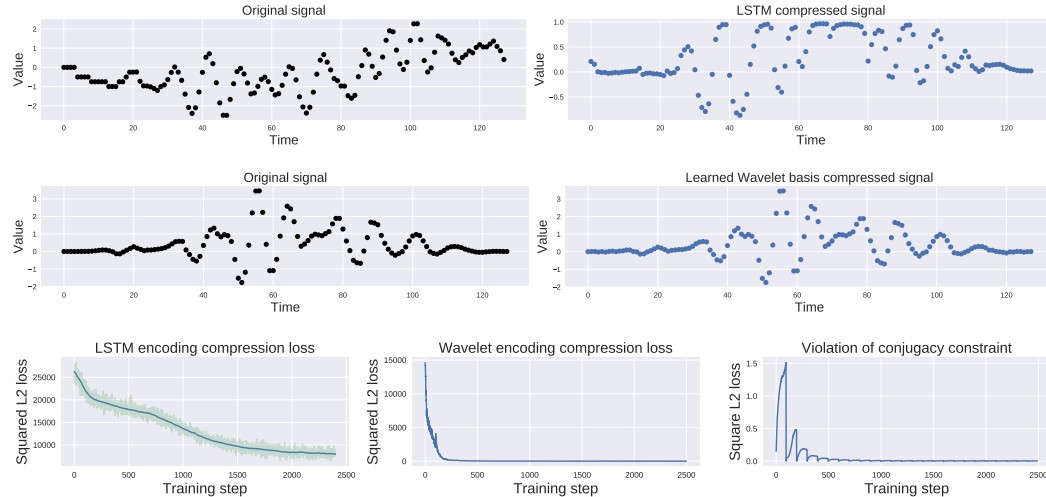

Figure 2: For a ratio of compression of 2, the learned wavelet basis auto-encoder outperforms an LSTM-based auto-encoder with the same number of parameters.

100 iterations. Figure 2 shows that this procedure progressively improves the randomly-initialized filters and significantly out-performs an LSTM-based auto-encoder model.

### 3.2 MULTI-SCALE VIDEO CLASSIFICATION

We further show that a wavelet representation can be composed with classical recurrent architecture (in regularly observed settings) to mitigate the effect of noisy data. This is particularly useful for LSTM networks Hochreiter & Schmidhuber (1997), since hyperbolic tangent layers tend to saturate in the presence of high-magnitude perturbations.

The YouTube-8M data-set contains millions of YouTube videos and their associated genres (Abu-El-Haija et al., 2016). Because the frames in each video are pre-featurized (i.e., a time series of featurized frames), models designed for this data-set must solely leverage the temporal structure in the data. In particular, the raw video feed is not available. A thorough description of the baselines we employ is available in Abu-El-Haija et al. (2016). This has enabled the authors of the paper to achieve state-of-the-art results in video classifications using a 2-layer LSTM model.

In our experiment, we train a similar model to learn on a multi-scale wavelet representation of data. This representation separates the original time series into $d$ scales, varying from fine to coarse. Each of the $d$ time series in this multi-scale representation are fed into a similar 2-layer LSTMs with $d^2$ times fewer parameters which results in a decrease of the total number of parameters in the recurrent layers by a factor of $d$. The outputs of each LSTM, are then concatenated before the final soft-max layer. We provide a model diagram detailing these components in the appendix. Our experimental results in Figure 3 indicate that this multi-scale representation greatly improves the performance of recurrent neural networks while using far fewer parameters.

### 3.3 MULTI-SCALE FINANCIAL FORECASTING

In 2015, an astounding medium volume of 40 million shares of AAPL (Apple Inc.) were traded each day. With the price of each share at approximately 100 USD, each 15-minute trading period represents an exchange of 142 million USD. Trades are highly irregular events characterized by an instantaneous exchange of shares between actors. Forecasting trade volume at a very fine resolution is essential in leveraging arbitrage opportunities. However, the noisy nature of financial markets makes this task incredibly challenging (Abergel et al., 2012).

Using AAPL price data from 2015, we train two neural networks: a standard LSTM and a wavelet transform network, to predict the next 15 minutes of trading given the 15 minutes that have just

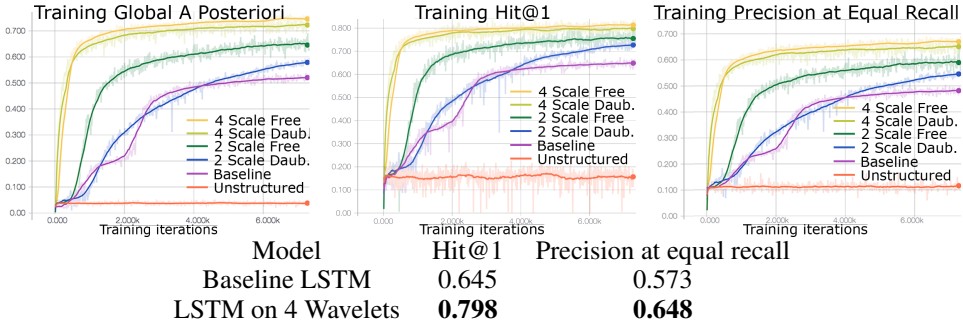

| Model | Hit@1 | Precision at equal recall |
|---|---|---|
| Baseline LSTM | 0.645 | 0.573 |
| LSTM on 4 Wavelets | **0.798** | **0.648** |

Figure 3: **Top:** An LSTM model trained on a wavelet-transformed YouTube-8M data-set achieves comparable results against the baseline while using half the number of parameters. **Bottom:** Table evaluating the wavelet-transformed LSTM model against the results from Abu-El-Haija et al. (2016) on held-out data.

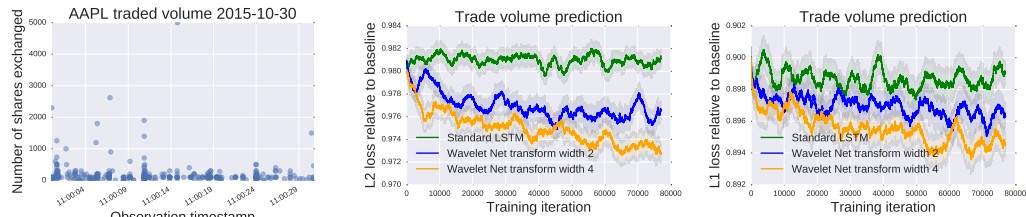

Figure 4: Predicting trade volume of a 15 minute window given the previous 15 minutes of observations. We evaluate the performance of neural networks against a simple averaging model. Even with $L_1$ regularization, the LSTM does not have the same predictive power as the neural network specified in Theorem 2.2 with $L = 2$ or $L = 4$.

elapsed. On average, the duration between time-stamps was 907ms (25th percentile: 200ms, median: 220ms, 75th percentile: 1800ms).

After the first scale projection onto a Haar wavelet basis (Mallat, 2008) is produced (with a characteristic resolution $\tau = 8$ seconds), both the wavelet transform network (with $M = 8$) and the LSTM make predictions with this first scale as input. Each model is evaluated by the $L_2$ loss against a baseline predicting a constant trading volume equal to the average over the previous $15$ observed minutes. Notice that in Figure 4, the LSTM struggles with the noisiness of the data, whereas the wavelet transform network is robust, and manages to improve the prediction performance by a half-percent. This half-percent represents 50 thousand USD of exchanged volume over a 15 minute period.

## 4 CONCLUSION

In this article, we analyze neural networks from a frame theoretic perspective. In doing so, we come to the conclusion that by considering time series as an irregularly observed continuous-time stochastic processes, we are better able to devise robust and efficient convolutional neural networks. By leveraging recent contributions to frame theory, we prove properties about non-linear frames that allow us to make guarantees over an entire class of convolutional neural networks. Particularly regarding their capacity to produce discrete representations of continuous time signals that are both injective and bi-Lipschitz. Moreover, we show that, under certain conditions, these properties *almost certainly* hold, even when the signal is irregularly observed in an event-driven manner. Finally, we show that bounded-output recurrent neural networks do not satisfy the sufficient conditions to yield non-linear frames.

This article is not limited to the theoretical statements it makes. In particular, we show that we can build a convolutional neural network that effectively computes a Discrete Wavelet Transform. The network's filters are dynamically learned while being constrained to produce outputs that preserve both orthogonality and the properties associated with non-linear frames. Our numerical experiments

on real-world prediction tasks further demonstrate the benefits of such neural networks. Notably, their ability to produce compact representations that allow for efficient learning on latent continuous-time stochastic processes.

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

CONVERGENCE OF WAVELET APPROXIMATIONS IN HOLDER FUNCTIONAL SPACES

We rely on Wavelet approximations (Mallat, 2008) to represent signals that are not band-limited. Given a certain Wavelet function $W \in L^2(\mathbb{R})$ and $W_{l,\tau} = \frac{1}{\sqrt{2^l}} W(\frac{\cdot - \tau}{2^l})$ a series of dilated and translated versions of $W$, a Wavelet approximation expresses a function $f \in L^2(\mathbb{R})$ as

$$f \simeq \mathrm{P}_W^L(f) \sum_{l=0}^{L-1} \sum_{\tau \in \mathbb{Z}} <f, W_{l,\tau}> W_{l,\tau}.$$

Under some conditions on $W$ (Mallat, 2008), the family $W_{l,\tau} = \frac{1}{\sqrt{2^l}} W(\frac{\cdot - \tau}{2^l})$ can be orthonormal and every function $f \in L^2(\mathbb{R})$ can be written in the limit as $f = \sum_{l \in \mathbb{Z}} \sum_{\tau \in \mathbb{Z}} <f, W_{l,\tau}> W_{l,\tau}$. A Wavelet function is defined as the high frequency mirror of a low frequency Scale function whose unit translations constitute a set of orthonormal atoms for a frame of $L^2(\mathbb{R})$ (i.e. a Riesz basis of $L^2(\mathbb{R})$).

In the following we consider functions with bounded support and restrict our study to functions defined on the interval $[0, 1]$ to simplify notations. A change of variable can immediately be employed to generalize the statements below to any bounded support function.

**Definition 4.1** *Holder space of $\alpha$ Lipschitz functions: A function $f \in L^2([0, 1])$ is uniformly $\alpha$ Lipschitz if there exists $M > 0$ such that for all $s$ in $[0, 1]$ there exists a polynomial $p_v$ of degree $\lfloor \alpha \rfloor$ such that*

$$\forall t \in [0, 1], |f(t) - p_v(t)| \le K|t - s|^\alpha.$$

In other words we consider functions defined on compacts that can be well approximated by polynomial splines and therefore have a certain degree of smoothness.

**Proposition 4.1** *Wavelet approximation of smooth functions with compact support: If $f \in L^2([0, 1])$ is $\alpha$ Lipschitz there exist $0 < m \le M$ such that the approximation error of the wavelet decomposition with wavelet function $W$ and scale function $S$ is*

$$\left\| f - \left( \sum_{l=L+1}^{\lfloor \log_2(N) \rfloor} \sum_{\tau=0}^{2^l-1} <f, W_{l,\tau}> W_{l,\tau} + \sum_{\tau=0}^{\lfloor \log_2(N) \rfloor-1} <f, S_{\lfloor \log_2(N) \rfloor,\tau}> S_{\lfloor \log_2(N) \rfloor,\tau} \right) \right\|_2^2 = O(2^{-N2\alpha})$$

The proposition above, proven in (Mallat, 2008) helps us examine how such an approximation affects the representations we employ.

## NEURAL ARCHITECTURES EMPLOYED IN THE VIDEO CLASSIFICATION EXPERIMENT

The present section of the appendix describes in detail the architectures employed in our Youtube8 video classification experiments. In Figure 5 we present the baseline architecture in the form of a block-diagram. In Figure 6 we expose our architectural choices which split the input into independent sub-spaces of representation that all correspond to different characteristic scales of variation of the input signal.

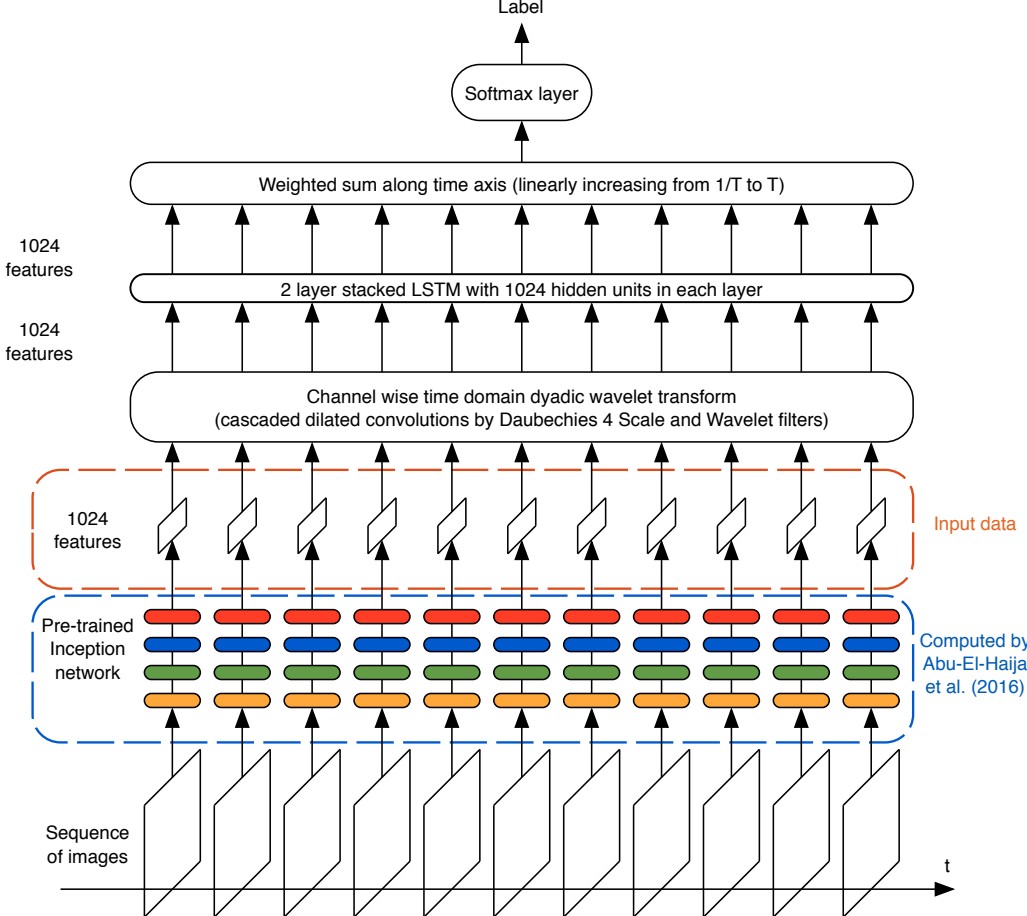

Figure 5: Baseline architecture from (Abu-El-Haija et al., 2016).

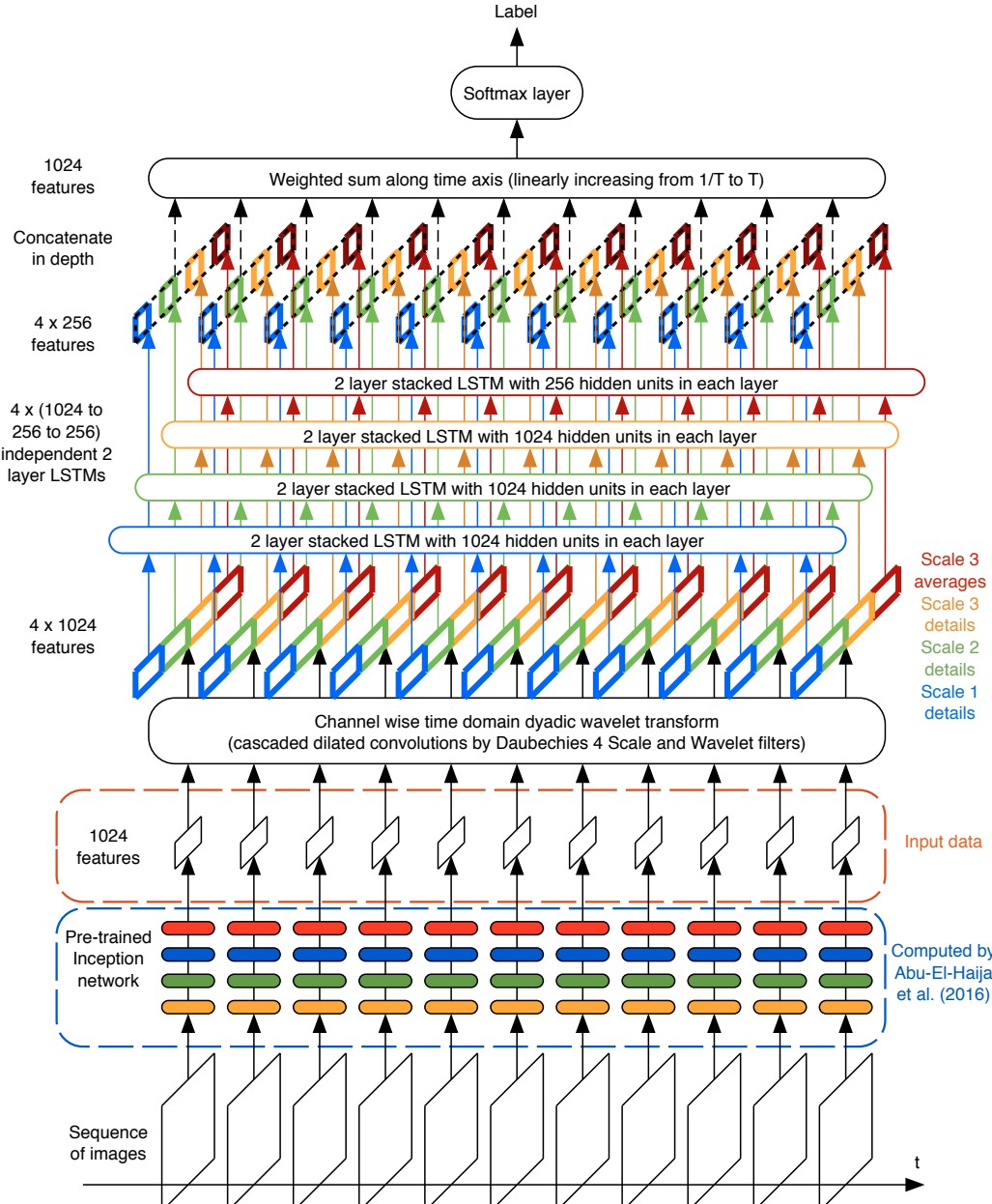

Figure 6: The architecture we propose for the Youtube video classification task that leverages a multi-resolution approximation computed by a wavelet convolution stack.

