# OpenReview forum: "Neural Networks for irregularly observed continuous-time Stochastic Processes"
_ICLR.cc/2018/Conference — Reject_

### Official Review · AnonReviewer1 · 2017-11-25
**The paper proves that a class of convolutional neural networks are nonlinear frames. Experiments demonstrate advantage of introducing CMF constraints on the filters in video classification and financial forecasting**

**Rating:** 5
**Confidence:** 5

**Review:**

The authors proved that convolutional neural networks with Leaky ReLU activation function are nonlinear frames, and similar results hold for non-uniformly sampled time-series as well. My main concern on this part is that theory is too rough and its link to the later part of the paper is weak. Although frames are stable representations, the ones with lower bound much smaller than the upper bound are close to unstable. That's why in classical applications of frames in signal and image processing tight frames are vastly preferred. Furthermore, the authors did not explicitly state the reliance of the lower frame bound on the parameter alpha in Leaky ReLU. It seems to me that the representation gets more unstable as alpha decreases, and the lower bound will be zero when ReLU is used.

In Section 2, the authors used CMF conditions to constraint filters which leads to a much more stable representation than in the previous section. The idea is very similar to previous work on data-driven tight frame (Cai et al. Applied & Computational Harmonic Analysis, vol. 37, no. 1, p. 89-105, 2014) and AdaFrame (Tai and E, arXiv:1507.04835). Furthermore, there has been multiple work on introducing tight-frame-like constraints to filters of convolutional neural networks (see for example Huang et al., arXiv:1710.02338). All of these work is not mentioned by the authors. Although the CMF constraints used by the authors seem new, the overall novelty is still weak in my opinion.

The experimental results are convincing, and the proposed architecture with wavelet-transform LSTM outperform the baseline model using standard LSTM. However, I am not familiar with the state-of-the-art models on the data sets used in the paper. Therefore, I am not certain whether the proposed method achieves state-of-the-art or not.

---

> ### Author Response · Authors · 2018-01-04
> **We thank the reviewer and have taken the remarks into account.**
>
> We thank the reviewer for the careful examination of the paper and apologize for the many typos that have hampered the reading.
> We we will now respond to the comments and remarks point by point.
>
> The reviewer’s remark on the role of alpha in Leaky ReLUs is quite accurate indeed. Although it is clear that a lower alpha means that our framing bounds will be less well conditioned, it is difficult to control the effect of the leaky part of the ReLU layer on conditioning when multiple such layers are employed. In particular, depending on the input the layers may or may not be in the leaky region of their support.
>
> As highlighted by the references given by the reviewer the tightness of a frame is of key importance when it comes to controlling the stability of the representation. In the present paper we only give properties on the smoothness of the representation because we explicitly consider a setting in which observations are observed irregularly and randomly through a Hawkes process. Our concern is therefore slightly different in that we attempt at tackling some properties of representations of randomly observed stochastic processes as observations go through a pipeline of non-linear operators.
>
> Again we thank the reviewer for helping us improve the paper.

---

### Official Review · AnonReviewer2 · 2017-11-25
**Motivation**

**Rating:** 5
**Confidence:** 4

**Review:**

Pros:
- combination of wavelets & CNN

Cons:
- lack of motivation

I am not sure to understand the motivation of good reconstruction/homeomorphism w.r.t. the numerical setting or combination with a CNN. (except for the first experiment) ; I give some comments section per section

Section 1:
Definition 1.1: why is it squared?
Definition 1.3: with this definition, "the framing constants" are not unique, so it should be "some framing constants"
There is a critical assumption to have an inverse, which is its stability. In particular, the ratio B/A is the quantity of interest. Indeed, a Gaussian filtering is Bi-Lipschitz-Invertible with this definition, yet, however it is quite hard to obtain the inverse which is not stable (and this is the reason why regularization is required in this inverse problem) Consequently, the assumption that CNNs are full rank does not really help in this setting(you can check the singular values). The conditioning is the good quantity to consider.

The Proposition 1.4 is trivial to prove, however I do not understand the following:
"With such vanishing gradients, it is possible to find series of inputsequences that diverge in l2(Z) while their outputs through the RNN are a Cauchy sequence"

How would you prove it or do you have some numerical experiments to do so?

Section 2:
The figure of the Theorem 2 is not really clear and could be improved. Furthermore, the 1x1 convolutions which follows the conjugate mirror filtering are not necessarily unitary.. This would require some additional constraints.

Subsection2.3:
The modulus is missing in the first sentence (on the fourier transform)

Section 3:
I find great the first experiment (which seems to indicate this particular problem is well conditioned). Nevertheless, the second experiment claims to improve the accuracy of the task while reducing the parameters, however it would be great to understand why there is this improvement. Similarly the last problem is better handled by the haar basis, is it because it permits the NN to learns to denoise or is it a conditioning issue? My guess is that it is because this basis sparsify the input signal, but it would require some additional experiments, in particular to understand how the NN uses it.

---

> ### Author Response · Authors · 2018-01-04
> **We thank the reviewer and have taken the remarks into account**
>
> We thank the reviewer for the careful examination of the paper and apologize for the many typos that have hampered the reading.
> We we will now respond to the comments and remarks point by point:
>
> Section1:
> Definition 1.1: framing conditions are generally expressed in terms of energy hence the square.
>
> Definition 1.3: framing constant are indeed not unique in our definition and the issue of lack of uniqueness has now been corrected. The conditioning number is indeed the core quantity of interest in the traditional setting of linear frames. As we delve into the non-linear setting we establish less stringent conditions that only attempt at guaranteeing homeomorphic properties.
> Proposition 1.4: the remark was a reference to the issues highlighted in the literature on RNNs concerning vanishing gradients. We now refer the reader to Bengio et al. 1993 in order to give the background of the remark.
>
> Section 2:
> We added the following remark at the end of 2.2: “A key point here is that the 1x1 convolutions operate in depth and not along the axis of time which preserves the properties of the Discrete Wavelet Transform.”
>
> Subsection 2.3:
> We thank the reviewer for having noticed the typographic error, it has been corrected.
>
> Section 3:
> The remarks of the reviewer about the need to better delineate the effects underlying the improvements we noticed highlight a key shortcoming of the experiments we have conducted. Our experimental setup is designed to provide evidence of gains in our representation. However, as with many evaluations of highly nonlinear deep neural networks it is difficult to resolve the precise gains due to individual changes.
>
> Again we thank the reviewer for helping us improve the paper.

---

### Official Review · AnonReviewer3 · 2017-11-27
**There are mistakes in the mathematical statements, and the motivations behind the work are not very clear. I do not think that this article is ready for publication.**

**Rating:** 2
**Confidence:** 3

**Review:**

Summary

This article considers neural networks over time-series, defined as a succession of convolutions and fully-connected layers with Leaky ReLU activations. The authors provide relatively general conditions for transformations described by such networks to admit a Lipschitz-continuous inverse. They extend these results to the case where the first layer is a convolution with irregular sampling. Finally, they show that the first convolutional filters can be chosen so as to represent a discrete wavelet transform, and provide some numerical experiments.


Main remarks

While the introduction seemed promising, and I enjoyed the writing style, I was disappointed with this article.

(1) There are many mistakes in the mathematical statements. First, in Theorem 1.1, I do not think that phi_L \circ ... \circ phi_1 \circ F is a non-linear frame, because I do not see why it should be of the form of Definition 1.2 (what would be the functions psi_n?). For the same reason, I also do not understand Theorem 1.2. In Proof 1.4, the line of equalities after « Also with the Plancherel formula » is, in my opinion, not true, because the L^2 norm of a product of functions is not the product of the L^2 norms of the functions. It also seems to me that Theorem 1.3, from [Benedetto, 1992], is incorrect: it is not the limit of t_n/n that must be larger than 2R, but the limit of N_n/n (with N_n the number of t_i's that belong to the interval [-n;n]), and there must probably be a compatibility condition between (t_n)_n and R_1, not only between (t_n)_n and R. In Proposition 1.6, I think that the equality should be a strict inequality. Additionally, I do not say that Proof 2.1 is not true, but the fact that the undersampling by a factor 2 does not prevent the operator from being a frame should be justified.

(2) The authors do not justify, in the introduction, why admitting a continuous inverse should be a crucial criterion of quality for the representation described by a neural network. Additionally, the existence of this continous inverse relies on the fact that the non-linearity that is used is a Leaky ReLU, which looks a bit like "cheating" to me, because the Lipschitz constant of the inverse of a Leaky ReLU, although finite, is large, so it seems to me that cascading several layers with Leaky ReLUs could encode a transformation with strictly positive, but still very poor frame bounds.

(3) I also do not understand why having "orthogonal outputs", as in Section 2, is really desirable; I think that it should be better justified. Also, there are probably other ways to achieve orthogonality than using wavelets in the first layer, so the fact that wavelets achieve orthogonality does not really justify why using wavelets in the first layer is a good choice, compared to other filters.

(4) I had understood in the introduction that the authors would explain how to define a (good) deep representation for data of the form (x_n)_{n\in\N}, where each x_n would be the value of a time series at instant t_n, with the t_n non-uniformly spaced. But all the representations considered in the article seem to be applicable to functions in L^2(\R) only (like in Theorem 1.4 and Theorem 2.2), and not to sequences (x_n)_{n\in\N}. There is something that I did not get here.


Minor remarks

- Fourth paragraph, third line: "this generalization frames"?
- Last paragraph before "Contributions & Organization": "that that".
- Paragraph about notations: it seems to me that what is defined as l^2(R) is denoted as l^2(Z) after the introduction.
- Last line of this paragraph: R^d_1 should be R^{d_1}, and R^d_2 R^{d_2}.
- I think "smooth" could be replaced by "continuous" (smoothness implies a notion of differentiability).
- Paragraph before Proposition 1.1: \sqrt{s} is not defined, and "is supported" should be "are supported".
- Theorem 1.1: the f_k should be phi_k.
- Definition 1.4: "piece-linear" -> "piecewise linear"?
- Lemma 1.2 and Proof 1.4: there are indices missing to \tilde h and \tilde g.
- Proof 1.4: "and finally" -> "And finally".
- Proof 1.5: I do not understand the grammatical structure of the second sentence.
- Proposition 1.4: the definition of a RNN is the same as definition 1.2 (except for the frame bounds); I do not see why such transformations should model RNNs.
- Paragraph before Proposition 1.5: "in,formation".
- Proposition 1.6: it should be said on which space the frame is injective.
- On page 8, "Lipschitz" is erroneously written (twice).
- Proposition 1.7: "ProjW,l"?
- Definition 2.1: in the "nested" property, I think that the inclusion should be the other way around.
- Before Theorem 2.1, the sentence "Such Riesz basis is proven" is unclear to me.
- Theorem 2.1: "filters convolution filters".
- I think the architecture described in Theorem 2.2 could be clarified; I am not exactly sure where all the arrows start from.
- First line of Subsection 2.3: ". is always" -> "is always".
- First paragraph of Subsection 3.2: "the the".
- Paragraph 3.2: could the previous algorithms developed for this dataset be described in slightly more detail? I also do not understand the meaning of "must solely leverage the temporal structure".
- I think that the section about numerical experiments could be slightly rewritten, so that the architecture used in each experiment is clearer. In Paragraph 3.2 in particular, I did not get why the architecture presented in Figure 6 has far fewer parameters than the one in Figure 5; it would help if the authors clearly precised how many parameters each layer contains.
- Conclusion: "we can to" -> "we can".
- Definition 4.1: p_v(s) -> p_v(t).

---

> ### Author Response · Authors · 2018-01-04
> **While we agree that there were several key typos, we think our claims are correct. We thank the reviewer and have taken the remarks into account.**
>
> We thank the reviewer for the careful examination of the paper and apologize for the many typos that have hampered the reading.
>
> Main remarks:
>
> (1)There are mistakes in the mathematical statements.
> While we agree that there were several key typos, we think our claims are correct.
>
> A) Theorem 1.1:
> F is a function from L2(R) to l2(Z) (which defines the psi_n) while phi_1 … phi_L are functions from l2(Z) to l2(Z). Therefore the overall composed operator is from L2(R) to l2(Z).
>
> B) Proof 1.4 presents a typographic error but remains true. Instead of an equality there is an inequality that stems from the fact that the L2 norm is algebraic and therefore the product of the norms is an upper bound to the norm of the product.
>
> C) Indeed, t_n / n is a typo. The ratio of interest is n/t_n and the constant is R_1.
>
> D) There is equality here because we defined R_1 > R and had the condition in theorem 1.3 depend on R.
>
> E) As explained in proof 2.1, the discrete wavelet transform preserve framing thanks to a careful down sampling scheme through a mirror filter bank. The critical conditions (in particular the mirroring filters) from the Mallat-Meyer theorem (which can be found in Mallat 2008)  on the filters guarantee that framing is preserved.
>
> (2) The article extends the study of homeomorphic properties from the linear to the non-linear case. In particular we show that the theory of frames which was mainly defined for linear operators can be employed to better understand non-linear functions.
> The leaky relus we employ have a leakiness factor of 0.1 and therefore their lipschitz constant is only 10.
>
> (3) Having orthogonal outputs is a sufficient condition to ensure that there are no linear level redundancies in the representation. Although other orthogonal transforms such as Fourier or Hadamard transforms could be employed, we rely on the general orthogonality of Wavelet basis.
>
> (4) As explained in the introduction of the article the input we consider is a function of L2(R) (continuous time object) and we study the impact of sequential sampling (observation is in l2(Z)) in the setting of non-linear operators.
> The very focus of the article is indeed the fact that most time series exist as continuous time objects (temperature, latent sentiment, location) but are only observed as discrete sequences. The article examines the consequences of discrete sampling on the representation of a continuous time latent process.
>
> Minor remarks
> Typographic errors have been corrected.
>
> - I think "smooth" could be replaced by "continuous" (smoothness implies a notion of differentiability).
> R: Here we ask for Lipschitz continuity which is stronger than continuity.
>
> - Paragraph before Proposition 1.1: \sqrt{s} is not defined, and "is supported" should be "are supported".
> R: The typographic errors have been corrected, sqrt(s) is now sqrt(delta t).
>
> - Proof 1.5: I do not understand the grammatical structure of the second sentence.
> R: “are also bi-Lipschitz” has been added at the end of the sentence.
>
> - Proposition 1.4: the definition of a RNN is the same as definition 1.2 (except for the frame bounds); I do not see why such transformations should model RNNs.
> R: An index has been added which was missing and now makes the difference between RNNs and non-linear frames explicit.
>
> - Proposition 1.6: it should be said on which space the frame is injective.
> R: We made the condition on the support of the Fourier transform of the functions of interest explicit.
>
> - Proposition 1.7: "ProjW,l"?
> R: The typographic error has been corrected. The W is now in index as intended while the scale l is presented as an exponent.
>
> - Definition 2.1: in the "nested" property, I think that the inclusion should be the other way around.
> R: The typographic error has been corrected.
>
> - Before Theorem 2.1, the sentence "Such Riesz basis is proven" is unclear to me.
> R: The typographic error has been corrected, “to exist” was missing.
>
> - Paragraph 3.2: could the previous algorithms developed for this dataset be described in slightly more detail? I also do not understand the meaning of "must solely leverage the temporal structure".
> R: In order to clarify our sentence, we added “In particular, the raw video feed is not available.” We also added “A thorough description of the baselines we employ is available in~\cite{abu2016youtube}.”
>
> - I think that the section about numerical experiments could be slightly rewritten, so that the architecture used in each experiment is clearer. In Paragraph 3.2 in particular, I did not get why the architecture presented in Figure 6 has far fewer parameters than the one in Figure 5; it would help if the authors clearly precised how many parameters each layer contains.
> R: We added: “which results in a decrease of the total number of parameters in the recurrent layers by a factor of d”.
>
> Again we thank the reviewer for helping us improve the paper.

---

> > ### Comment · AnonReviewer3 · 2018-01-08
> > **Thank you for the corrections; I do not agree with all of them.**
> >
> > Dear authors,
> >
> > Thank you for your reply, and thank you for correcting various typos. I however do not agree with all your corrections.
> >
> > Theorem 1.1:
> > I agree that the composed operator is from L2(R) to l2(Z). However, not all operators from L2(R) to l2(Z) are of the form f -> (psi_n(\scal{f}{S_n}))_n, for some family of elements S_n in L2(R) and some family of operators psi_n, as in Definition 1.2.
> >
> > Proof 1.4:
> > The product of L2 norms of functions defined over [0;1] is not an upper bound to the norm of the product, and the inequality ||h* x||_2 <= ||h||_2 ||x||_2 is not true for all h and x (an easy counter-example is h=x=\delta_0+\delta_1).
> >
> > Proposition 1.6:
> > I still do not get it. if R = 1/2 mu/(1-int \phi), then, from Propositon 1.5, N_t/t goes to 2R when t goes to infinity. Then n/t_n also goes to 2R when n goes to infinity, and since R1 > R, the limit of n/t_n is not larger than 2R1 as required by Theorem 1.3.
> >
> > About the motivations, I still think that they should be developed. Why is it important for a neural network to admit a continuous inverse? And about orthogonal outputs, I am not sure that getting rid of redundancy is necessary, or desirable; I think it should be discussed.

---

### Decision · Program_Chairs · 2018-01-29
**ICLR 2018 Conference Acceptance Decision**

**Decision:**

Reject

**Comment:**

The scores were not favorable: 5,5,2. R2 felt the motivation of the paper was inadequate. R3 raised numerous technical points, some of which were addressed in the rebuttal, but not all. R3 continues to have issue with some of the results. The AC agrees with R3's concerns and feels that the paper cannot be accepted in its current form.